# Sarcospan Deficiency Increases Oxidative Stress and Arrhythmias in Hearts after Acute Ischemia-Reperfusion Injury

**DOI:** 10.3390/ijms241411868

**Published:** 2023-07-24

**Authors:** Hyun Seok Hwang, Aida Rahimi Kahmini, Julia Prascak, Alexis Cejas-Carbonell, Isela C. Valera, Samantha Champion, Mikayla Corrigan, Florence Mumbi, Michelle S. Parvatiyar

**Affiliations:** Department of Nutrition and Integrative Physiology, Florida State University, 107 Chieftan Way, Biomedical Research Facility, Tallahassee, FL 32306-1490, USA

**Keywords:** arrhythmia, ischemia-reperfusion injury, oxidative stress, sarcospan

## Abstract

The protein sarcospan (SSPN) is an integral member of the dystrophin-glycoprotein complex (DGC) and has been shown to be important in the heart during the development and the response to acute stress. In this study, we investigated the role of SSPN in the cardiac response to acute ischemia-reperfusion (IR) injury in SSPN-deficient (SSPN^−/−^) mice. First, the hemodynamic response of SSPN^−/−^ mice was tested and was similar to SSPN^+/+^ (wild-type) mice after isoproterenol injection. Using the in situ Langendorff perfusion method, SSPN^−/−^ hearts were subjected to IR injury and found to have increased infarct size and arrhythmia susceptibility compared to SSPN^+/+^. Ca^2+^ handling was assessed in single cardiomyocytes and diastolic Ca^2+^ levels were increased after acute β-AR stimulation in SSPN^+/+^ but not SSPN^−/−^. It was also found that SSPN^−/−^ cardiomyocytes had reduced Ca^2+^ SR content compared to SSPN^+/+^ but similar SR Ca^2+^ release. Next, we used qRT-PCR to examine gene expression of Ca^2+^ handling proteins after acute IR injury. SSPN^−/−^ hearts showed a significant decrease in L-type Ca^2+^ channels and a significant increase in Ca^2+^ release channel (RyR2) expression. Interestingly, under oxidizing conditions reminiscent of IR, SSPN^−/−^ cardiomyocytes, had increased H_2_O_2_-induced reactive oxygen species production compared to SSPN^+/+^. Examination of oxidative stress proteins indicated that NADPH oxidase 4 and oxidized CAMKII were increased in SSPN^−/−^ hearts after acute IR injury. These results suggest that increased arrhythmia susceptibility in SSPN^−/−^ hearts post-IR injury may arise from alterations in Ca^2+^ handling and a reduced capacity to regulate oxidative stress pathways.

## 1. Introduction

Pathogenic variants in the dystrophin-glycoprotein complex (DGC) lead to cardiac and skeletal muscle disorders [1]. The DGC consists of the peripheral membrane protein α-dystroglycan, integral membrane protein β-dystroglycan, transmembrane sarcoglycans (α, β, δ, γ), sarcospan (SSPN) and dystrophin. Duchenne muscular dystrophy (DMD) is caused by pathogenic variants in the *DYS1* gene leading to dystrophin loss and the *mdx* mouse model was developed to study the disease [2,3]. Loss of dystrophin and SGs from the sarcolemma of striated muscles causes membrane instability and contraction-induced injury [4,5,6,7]. Subsequently, the development of microtears allows for dysregulated calcium ion entry into the muscle [5,6].

The protein SSPN has been shown to be important in maintaining cardiac and skeletal muscle function and integrity [8,9]. Unlike the other DGC subunits, it remains unclear whether SSPN has a role in pathogenesis. However, there was a recent genome-wide association study (GWAS) that identified *SSPN* as a candidate gene for atrial fibrillation (AF) [10]. This study was the first report implicating SSPN in arrhythmogenesis. The origin of AF is still incompletely understood and investigations into the hereditary origins of the disease are ongoing. Additional variants in cytoskeletal proteins have been implicated in AF and their pathogenesis linked to the disruption of cardiomyocyte structures [11,12,13]. Since SSPN has an important role in sarcolemma stabilization, variants in the gene may increase arrhythmia susceptibility by altering membrane stability and function/localization of membrane proteins involved with ion exchange and/or conduction. The previous studies showed that chronic β-adrenergic challenge of SSPN^−/−^ mice resulted in a blunted cardiac functional response compared to wild type mice [14]. Since β-adrenergic receptor expression was maintained—the lack of functional responsiveness may be attributed to other mechanisms that include altered cardiomyocyte Ca^2+^ handling. Overall, this study is aimed at understanding the normal functions of SSPN in the developing and adult heart that protect the heart from injury and arrhythmias [15,16].

To uncover arrhythmogenic triggers for the SSPN^−/−^ myocardium, we utilized several arrhythmia-inducing stressors, including β-adrenergic stimulation and ischemia-reperfusion (IR) injury. We also compared the impact of SSPN loss to that of its closely associated proteins. One protein that SSPN makes significant associations with is β1D integrin [14,17,18] and when SSPN is absent in the heart β1D integrin appears less tightly associated with the DGC [14]. Integrin proteins are heterodimeric transmembrane receptors that are expressed in cardiomyocytes and all other cells [19]. Integrin heterodimers have been found in intercalated discs (ICDs) that connect cardiomyocytes end-to-end and mediate mechanical and electrical coupling in the heart. Furthermore, β1D integrin is downregulated in hearts after a myocardial infarction and integrin-deficient animal models have been found more susceptible to ischemic injury [20]. The loss of β1D integrin may also contribute to a decline in heart function due to abrogated communication between cardiomyocytes and the extracellular matrix (ECM) [21]. Since SSPN expression impacts β1D integrin sarcolemma abundance and its associations, we were interested in whether its deletion would render the myocardium more susceptible to injury and arrhythmias.

This is the first study examining the role of SSPN in cardiomyocyte Ca^2+^ handling and response to IR injury. To investigate a role for SSPN in arrhythmia and acute IR injury, we examined ex vivo Langendorff perfused SSPN^−/−^ hearts to assess the intrinsic mechanisms governing electrical activity and conduction independent of autonomic nervous regulation.

## 2. Results

### 2.1. Determining the Acute Hemodynamic Response of Sarcospan-Deficient Hearts to Cardiac Stress Testing

Our initial experiments were designed to assess the physiological response in SSPN^−/−^ mice compared to age-matched SSPN^+/+^ controls. In Figure 1A a representative electrocardiogram (ECG) trace is shown prior to and after intraperitoneal (i.p.) administration of isoproterenol challenge to SSPN^+/+^ or SSPN^−/−^ mice. After the administration of the isoproterenol bolus, both SSPN^+/+^ and SSPN^−/−^ mice exhibited a similar increase in heart rate, recorded as beats per minute (bpm), increasing from baseline values of ~450 bpm to ~550 bpm for SSPN^+/+^ and ~600 bpm for SSPN^−/−^ as shown in Figure 1B. The measurements of heart rate variability (HRV) are shown in Figure 1C in SSPN^+/+^ or SSPN^−/−^ mice before and after the isoproterenol treatment indicate a similar decrease in HRV during stress testing. Next, we assessed the in vivo cardiac contractility in mice using echocardiography under 2% isoflurane with 100% O_2_. The SSPN^−/−^ mice displayed similar left ventricular fractional shortening (LV FS, %) and isovolumic relaxation time (IVRT, msec) compared to SSPN^+/+^ mice (Table 1). Overall, these studies indicate that SSPN-deficiency does not affect cardiac contractility at rest, and this does not impact the hemodynamic response of mice to acute β-adrenergic agonist administration.

### 2.2. Assessment of the Sarcospan-Deficient Cardiac Response to Acute Ischemia-Reperfusion Injury

To determine whether the SSPN^−/−^ myocardium is more susceptible to acute ischemia-reperfusion (IR) injury and the development of arrhythmias, we subjected ex vivo perfused SSPN^+/+^ and SSPN^−/−^ hearts to IR injury, according to the schematic shown in Figure 2A. During the reperfusion phase the intrinsic sinus atrial node pacemaker activity can be determined in Langendorff perfused hearts. Overall, there were no significant alterations in the sinus heart rate between SSPN^+/+^ and SSPN^−/−^ perfused hearts. Post-IR arrhythmia risk was assessed using an ECG during perfusion and the number of premature ventricular contractions (PVCs) per minute after IR injury were found to be significantly increased in SSPN^−/−^ compared to SSPN^+/+^ hearts.

### 2.3. Ca^2+^-Handling Measurements in Sarcospan-Deficient Cardiomyocytes after Acute β-Adrenergic Stimulation

To investigate the underlying factors contributing to the heightened risk of arrhythmias in the SSPN^−/−^ myocardium after acute IR injury cardiomyocytes were isolated from SSPN^+/+^ and SSPN^−/−^ hearts to assess perturbations in cellular Ca^2+^ handling. Cellular Ca^2+^ handling was reported in Fura-2 AM-loaded cardiomyocytes after 10 s of electrical stimulation under baseline conditions and after acute exposure to 1 μM isoproterenol (Iso). In Figure 3A, SSPN^+/+^ and SSPN^−/−^ cardiomyocytes exhibited similar F_ratio,_ at baseline indicating similar levels of diastolic Ca^2+^.However, after acute Iso exposure SSPN^+/+^ cardiomyocytes exhibited a significantly increased F_ratio._ This suggested that there was an increase in diastolic Ca^2+^ in SSPN^+/+^ cardiomyocytes; however, it remained unchanged in SSPN^−/−^ cardiomyocytes. In Figure 3B the Ca^2+^ transient is shown and reports the peak amplitude of the Ca^2+^ transient under resting (-Iso) and stimulated conditions (+Iso). Under resting conditions, the Ca^2+^ T height was significantly lower in SSPN^−/−^ compared to SSPN^+/+^ cardiomyocytes. Overall, there were no differences in the peak amplitude between SSPN^+/+^ and SSPN^−/−^ cardiomyocytes upon acute Iso exposure.

The length of time it takes the peak Ca^2+^ transient height to decay is reported in Figure 3C as Ca^2+^ T decay. A delay in Ca^2+^ T decay suggests action potential prolongation, which were significantly longer under stimulated conditions (+Iso), while there was also a trend toward delayed Ca^2+^ T decay under resting conditions. This provides important information about the balance between inward I_Ca_ and outward I_k_ currents; therefore, the prolonged Ca^2+^ transients seen in SSPN^−/−^ may increase the risk of early afterdepolarizations, leading to PVCs under stress conditions. In Figure 3D, the SR Ca^2+^ T height was measured after caffeine-induced SR Ca^2+^ release to estimate the SR Ca^2+^ content. The F_ratio_ increased after Iso treatment, indicating higher diastolic Ca^2+^ (greater Ca^2+^ SR release) for both SSPN^+/+^ and SSPN^−/−^ cardiomyocytes. However, SSPN^−/−^ cardiomyocytes had significantly lower SR Ca^2+^ T height compared to SSPN^+/+^ after Iso treatment. After caffeine administration, Iso treatment SR Ca^2+^ content was significantly higher for both SSPN^+/+^ and SSPN^−/−^ cardiomyocytes indicated by F_ratio_. To better understand the functional SR Ca^2+^ release, the Ca^2+^ T height was normalized to caffeine induced SR Ca^2+^ T height, Figure 3E indicating an overall similar significant fractional release (FR, %) between SSPN^+/+^ and SSPN^−/−^ under both conditions tested.

### 2.4. Alterations in Ca^2+^ Regulatory Protein Gene Expression in SSPN-Deficient-Hearts after Acute IR Injury

To assess acute changes in gene expression, qRT-PCR was utilized to assess the rapid regulatory responses of key genes responsible for modulating Ca^2+^ release and cardiac rhythm in response to IR injury. The expression of β-adrenergic receptor 1 (β-AR1) appeared unchanged between the groups after acute IR injury as shown in Figure 4A. After Iso treatment, β-AR1 expression was unchanged in both SSPN^−/−^ and SSPN^+/+^ hearts. Whereas expression of the plasma membrane Ca^2+^ handling protein L-type Ca^2+^ channel (LTCC) was increased in SSPN^+/+^ hearts upon IR injury with a concomitant decrease in in SSPN^−/−^ hearts as indicated in Figure 4B. The trends between Na^+^/Ca^2+^ exchanger (NCX) and RyR2 expression were similar; however, RyR2 expression was significantly increased while NCX expression trended higher in SSPN^−/−^ hearts as shown in Figure 4C and 4D. However, SERCA2 expression was unchanged between groups while after acute IR its regulator phospholamban (PLN) was significantly decreased in SSPN^+/+^ hearts as shown in Figure 4E,F. This indirectly suggests increased Ca^2+^ uptake through SERCA2.

### 2.5. Response of SSPN-Deficient Cardiomyocytes to Oxidative Stress Conditions

Since perfused SSPN^−/−^ hearts exhibited larger infarct size in response to acute IR injury, we assessed whether increased reactive oxygen species (ROS) production under oxidative stress could be a contributing factor to heightened myocardial damage in SSPN^−/−^ hearts. To assess this, in Figure 5A, primary SSPN^+/+^ and SSPN^−/−^ cardiomyocytes were loaded with fluorogenic dye DCFDA/H2DCFDA to obtain a quantitative assessment of ROS produced in response to cellular oxidants mimicked by H_2_O_2_ exposure. In the left panel of Figure 5A an untreated cardiomyocyte is shown, whereas in the right panel images of SSPN^+/+^ and SSPN^−/−^ cardiomyocytes exposed to oxidizing conditions for 30 min. In Figure 5B, the relative ROS production was quantified by assessing fluorescence in SSPN^+/+^ and SSPN^−/−^ cardiomyocytes due to the increased oxygen species. It is evident from these measurements that SSPN^−/−^ cardiomyocytes produced greater amounts of oxidant-induced excessive oxygen species ROS compared to SSPN^+/+^.

### 2.6. Assessment of Stress Responsive Protein Expression in SSPN-Deficient Hearts after Acute IR Injury

After acute IR injury the SSPN^+/+^ and SSPN^−/−^ perfused hearts were assessed for alterations in protein expression to better understand immediate alterations in protein stress responsive pathways. In Figure 6, representative immunoblot results are shown for the stress-responsive proteins atrial natriuretic peptide (ANP), NADPH oxidase 4 (NOX4), oxidized calcium/calmodulin-dependent protein kinase II (oxi-CAMKII) with glyceraldehyde-3-phosphate dehydrogenase (GAPDH) shown as the loading control. In Figure 6A the cardiac stress marker ANP was found to be unaltered between the groups. Alterations in ANP reflect an increase in cardiac load and overall cardiac stress and it was anticipated that it would be unchanged in our study. While an increase in NOX4 expression is well documented after IR injury in the myocardium, in Figure 6B it appears that SSPN^−/−^ hearts undergo a rapid antioxidant response since NOX4 protein levels increased upon acute IR injury. Furthermore, excessive ROS in the SSPN^−/−^ cytosol after IR injury increased oxidized CAMKII/total CAMKII as shown in Figure 6C. Increased levels of oxidized CAMKII can lead to dysregulated phosphorylation of cardiac Ca^2+^ release and uptake channels (RyR2 or SERCA2) [22].

## 3. Discussion

As a tetraspanin-like protein, SSPN participates in stabilizing protein-protein interactions between important sarcolemma adhesion complexes, including the DGC, UGC and integrin complexes [18,23,24]. Tetraspanin proteins function as molecular scaffolds that organize tetraspanin enriched microdomains (TEMs) that include interactions with other transmembrane proteins, tetraspanins and additional proteins on the membrane surface [25]. The previous studies in skeletal muscle suggest that SSPN plays a role in regulating the crosstalk between these three adhesion glycoprotein complexes as well as influencing integrin expression [26,27]. While SSPN clearly has an important stabilizing role for adhesion complexes at the cell surface [14]—tetraspanin proteins are well known to regulate adhesion receptors—most prominently integrin proteins [28]. In this study we addressed several questions: (1) are SSPN^−/−^ hearts responsive to an acute β-adrenergic stimulus? (2) Does SSPN-deficiency render the myocardium more susceptible to acute IR injury? (3) Does acute IR injury reveal arrhythmogenesis in SSPN^−/−^ hearts?

Acute stress testing was initially performed using isoproterenol (Iso) to assess the hemodynamic response of SSPN^+/+^ and SSPN^−/−^ hearts. SSPN^−/−^ hearts were responsive to acute β-adrenergic challenges with similar increases in heart rate (bpm) and decreases in heart rate variability as the SSPN^+/+^ hearts (Figure 1). As SSPN^−/−^ hearts did not reveal arrhythmogenesis in response to acute β-adrenergic stress we examined additional stressors. Since β1D integrin deficiency increases myocardial susceptibility to IR injury we reasoned that the loss of the integrin-associated protein SSPN may also increase myocardial risk. Multiple β1 integrin mouse models exist—cardiomyocyte-specific β1 integrin knockout hearts have increased vulnerability to IR injury [20], global knockout β1 integrin-deficient mice have reduced myocardial function after myocardial infarction [29], and cardiomyocyte specific overexpression of α7β1D integrins protect against IR injury [20]. Similar to β1D integrin-deficient mouse models SSPN^−/−^ mice were found to be more susceptible to IR injury.

In Figure 2, ex vivo hearts were monitored using ECG after IR injury and SSPN^−/−^ hearts were found to have increased incidence of PVCs, and, therefore, an increased risk of arrhythmias compared to SSPN^+/+^ hearts. After acute IR injury, the SSPN^−/−^ myocardium exhibited larger infarcted areas and more widespread damage. This suggests that SSPN affords global protection to cells throughout the heart. The increased vulnerability of the SSPN^−/−^ myocardium to acute IR can be attributed to a variety of factors, including reduced localization of dystrophin at the sarcolemma [14], altered interactions with β1D-integrin or other crucial membrane proteins [14] and the potential disruption of the submembrane cytoskeletal network. 

Perturbations to key pathways can increase cell death in response to acute IR injury; therefore, we examined alterations in key Ca^2+^ handling and redox proteins. Studies in the *mdx* mouse model are highly informative regarding the impact of complete dystrophin deficiency, destabilization of the DGC and perturbations in Ca^2+^ handling. Studies in *mdx* cardiomyocytes show that they have reduced resting sarcomere lengths and increased passive tension, which contributes to poor passive compliance [30]. Examining cardiomyocytes with deletion of other DGC proteins provides context for our study. The sarcoglycans (SGs) form subcomplexes with SSPN (SSPN-SG), cardiomyocytes obtained from mutant delta-sarcoglycan (δ-SG) mice presenting with dilated cardiomyopathy (DCM) exhibit stretch-induced defects in plasma stability [4]. Additional *mdx* cardiomyocyte experiments by Viola et al. suggest that the absence of dystrophin disrupts the normal cytoskeletal architecture, affecting LTCC activation and superoxide generation [31]. Whereas SSPN deletion does not appear to destabilize the sarcolemma, though it reduces the abundance of other DGC components in skeletal [32] and cardiac [14] muscle since neither show a significant uptake of Evan’s Blue Dye, an in vivo dye tracer. However, this has not been tested under acute stress conditions, such as IR injury.

The reduction of dystrophin in SSPN^−/−^ cardiomyocytes may diminish the “shock absorbing” capacity of the sarcolemma by weakening the connection between the ECM and intracellular actin and/or perturb its scaffolding function [33,34]. This could alter the localization and function of the membrane proteins involved in Ca^2+^ handling and redox homeostasis. Relatedly, the absence of dystrophin in DMD leads to mis-localization and abnormal expression/activity of Ca^2+^ handling and oxidative stress responsive proteins [34,35,36,37]. Since the SSPN^−/−^ myocardium does not exhibit overt signs of membrane instability—the increased vulnerability of the SSPN^−/−^ myocardium to IR injury could be attributed to aberrant cardiomyocyte Ca^2+^ entry or a reduction in submembrane scaffolding orchestrated by dystrophin and/or SSPN.

In this study, we examine alterations in Ca^2+^ handling in SSPN^−/−^ cardiomyocytes to better understand their cellular phenotype compared to the dystrophic cellular phenotype. Our results indicate that mechanisms leading to cytosolic Ca^2+^ overload in *mdx* cardiomyocytes may not be substantially activated in SSPN^−/−^ cardiomyocytes. To assess this, we examined intracellular Ca^2+^ handling in SSPN^−/−^ cardiomyocytes after exposure to acute isoproterenol regimen. We found that in contrast to chronic in vivo β-adrenergic stimulation of SSPN-deficient hearts, isolated SSPN^−/−^ cardiomyocytes remained responsive to acute isoproterenol exposure [14], see Figure 3.

To maintain normal function, cardiomyocytes undergo infinite adaptations to Ca^2+^ entry and efflux across the cell membrane to achieve a given average systolic and diastolic Ca^2+^ level. Stress induction regimens using acute β-adrenergic stimulation, provide information on the responsive balance of stress and adaptations of excitation-contraction coupling (ECC) [38] to uncover subthreshold differences in SSPN^−/−^ cardiomyocyte Ca^2+^ handling. Our Ca^2+^ handling data indicated that SSPN^−/−^ cardiomyocytes maintained responsiveness to acute isoproterenol exposure. Overall, isoproterenol exposure activates PKA through β-adrenergic receptors and increases the amount of SR Ca^2+^ release through increased Ca^2+^ current (*I*_Ca_) and SR Ca^2+^ contents [39]. Furthermore, faster release kinetics, due to β-adrenergic stimulation is mainly due to the phosphorylation of RyR2 [38]. SR Ca^2+^ height estimates the SR Ca^2+^ content of cardiomyocytes since caffeine triggers Ca^2+^ release by reducing the RyR2 activation threshold for luminal but not cytosolic Ca^2+^ activation [40]. SSPN^−/−^ cardiomyocytes had alterations in these two parameters (diastolic Ca^2+^ and SR Ca^2+^ content) revealing reduced Ca^2+^ SR content under two conditions: β-adrenergic and caffeine stimulation. After β-adrenergic stimulation, both the diastolic Ca^2+^ levels and SR Ca^2+^ contents were lower, which may be explained by the reduced SR Ca^2+^ release accompanied by the lower SR Ca^2+^ content.

It has been shown that transient alterations in Ca^2+^ handling gene expression occur upon acute IR injury [39]. These changes precede alterations in protein levels, protein turnover or protein activation but rather are fast-acting adaptations in response to rapid changes in cellular homeostasis. Furthermore, modifications to the epigenome occur rapidly in response to environmental signals [41]. Therefore, the 1.5-h IR protocol may be able to capture rapid changes in Ca^2+^ handling gene expression and protein levels. The initial response of healthy cardiac myocytes to electrical stimulation is an increase in intracellular Ca^2+^. This initially occurs as a small influx of Ca^2+^ through LTCC that triggers a larger Ca^2+^ release from the SR through RyR2 [42]. If loss of SSPN affects the organization or stability of major cell membrane Ca^2+^ handling proteins essential to ECC we anticipated affected proteins may exhibit transient gene expression changes in response to acute IR injury. The most striking rapid alteration in gene expression in SSPN^−/−^ hearts after acute IR injury was a reduction in LTCC and an increase in RyR2 expression see Figure 4. This is intriguing as both RyR2 and LTCC have important regulatory roles in maintaining cardiomyocyte Ca^2+^ homeostasis. Our initial findings suggest that tight transcriptional regulatory mechanisms may compensate for cellular perturbations that exist within SSPN^−/−^ cardiomyocytes.

The scaffold role of dystrophin was demonstrated and the loss of SSPN may potentially influence these processes by altering the activity/sarcolemma localization of proteins regulating redox homeostasis. Excessive ROS production that outpaces the antioxidant capacity of the heart contributes to myocardial damage in ischemic heart disease and heart failure [43]. Since SSPN^−/−^ hearts exhibited widespread damage after acute IR injury, we were interested in whether the antioxidative capacity of the SSPN^−/−^ myocardium was compromised. To assess this, we exposed SSPN^−/−^ cardiomyocytes to oxidizing conditions and measured the production of ROS in response to H_2_O_2_, see Figure 5. Therefore, the increase in H_2_O_2_-induced ROS production in SSPN^−/−^ cardiomyocytes compared to SSPN^+/+^ may be a contributing factor to the increased susceptibility of the SSPN^−/−^ myocardium to IR injury. Ca^2+^ overload also leads to increased ROS in the heart; however, the SSPN^−/−^ cardiomyocyte Ca^2+^ handling data presented here does not suggest major perturbations in Ca^2+^ regulation. Therefore, it is more likely that SSPN deficiency has a greater impact on the antioxidant mechanisms within the heart.

To address the finding that SSPN^−/−^ cardiomyocytes produce higher levels of ROS under oxidative conditions, we examined the protein expression of key oxidative stress markers in SSPN^−/−^ hearts after acute IR injury. This data is shown in Figure 6. In the heart, reduced nicotinamide adenine dinucleotide phosphate (NADPH) oxidases (NOX) 2 and 4 are enzymes that produce O^2-^ or H_2_O_2_ and are the major source of ROS. The suppression of either NOX2 or NOX4 has been demonstrated to reduce ROS and protect against IR injury [44]. Our study found that NOX4 expression was increased in SSPN^−/−^ hearts after IR injury. The previous studies of DMD-associated cardiomyopathy found increased induction of NOX4 in the diseased hearts associated with increased fibrosis and functional decline [45]. In response to IR injury, increased NOX4 expression is expected to heighten oxidative stress and contribute to transient oxidative stress-induced long-term facilitation of the LTCC current. This will further affect SR Ca^2+^ regulation in cardiomyocytes and activate CAMKII via oxidation-dependent mechanisms. Remodeling of the cardiac action potential by higher baseline oxidative stress-induced cellular memory may lead to pathological processes, increasing the susceptibility to cardiac arrhythmia.

## 4. Conclusions

Our findings suggest that SSPN deficiency increases myocardial risk upon IR injury by enhancing pathways that increase oxidative stress. This can cause increased damage to heart tissue under adverse conditions and affect systolic Ca^2+^ release. These alterations in SSPN^−/−^ hearts can contribute to increased number of premature ventricular contractions and altered propagation of signals throughout the heart leading to an increased risk of arrhythmias.

## 5. Materials and Methods

All animal protocols used in this study conformed to the Guiding Principles in the Care and Use of Animals of the American Physiological Society and were approved by The University Animal Care and Use Committee (IACUC Approval #: PROTO202100037) at Florida State University. The sspntm1Kcam (SSPN-deficient) mice were purchased from The Jackson Laboratory and developed by the Campbell laboratory [32] these mice were backcrossed on C57BL6/J and housed in groups of 4–5, fed a standard laboratory diet 5001 (PMI Nutrition International), provided with water ad libitum and maintained on a 12:12 h light-dark cycle. The study used a total of thirty-seven mice, which were divided into two groups based on their genotype (SSPN^+/+^ and SSPN^−/−^) and their age range was between 4 and 8 months. Differences between the groups were assessed using one-way ANOVA; whereas, the incidence of arrhythmia was compared using the non-parametric Mann-Whitney test.

## 6. Echocardiography

Echocardiography was performed under 2% of Isoflurane with 100% O_2_ for 15 min [46]. The echocardiographic parameters were measured from 3 consecutive beats and averaged. The Sonos 5500 system, with a linear transducer in 15 MHz modes, was used to obtain the standard two-dimensional (2D) short axis views. A 2D guided M-mode was performed on the left ventricle (LV) at the tip of the mitral leaflets and through the center of the LV cavity. The LV end-systolic and end-diastolic internal diameters, LVIDS and LVIDD, respectively, and the LV wall thickness were measured for each animal using the M-mode image in a blinded fashion. The LV fractional shortening was calculated as (LVIDD-LVIDS)/LVIDD.

## 7. Real-Time PCR and mRNA Expression

Total RNA was isolated from heart tissue using Trizol reagent (Life Technologies, Grand Island, NY, USA). For cDNA synthesis the qPCRBIO cDNA Synthesis Kit (PCR Biosystems) was used and each reaction contained 0.4 μg total RNA. The expression of mRNA was analyzed by Real-time quantitative PCR using 2X qPCRBIOSyGreen Mix (PCR BIOsystems) in QuantStudio 3 (Applied Biosystems). β-actin was used to normalize the mRNA. The relative expression was determined using the ΔΔCt method.

The sequences of the primers utilized are as follows:

β-AR1-F: 5′-CTC ATC GTG GTG GGT AAC GTG-3′ and R: 5′-ACA CAC AGC ACA TCT ACC GAA-3′;

LTCC-F: 5′-ATG AAA ACA CGA GGA TGT ACG TT-3′ and R: 5′-ACT GAC GGT AGA GAT GGT TGC-3′;

NCX1-F: 5′-AAA GAG TGC AGT TTC TCC CTT G-3′ and R: 5′-TGA AGC CAC CTT TCA ATC CTC-3′;

RyR2-F: 5′-ACG GCG ACC ATC CAC AAA G-3′ and R: 5′-AAA GTC TGT TGC CAA ATC CTT CT-3′;

SERCA2-F: 5′-GAG AAC GCT CAC ACA CAA AGA CC-3′ and R: 5′-CAA TTC GTT GGA GCC CCA T-3′;

PLN-F: 5′-AAA GTG CAA TAC CTC ACT CGC-3′ and R: 5′-GGC ATT TCA ATA GTG GAG GCT C-3′; and

β-actin-F: 5′-GGG TGT ATT CCC CTC CAT CG-3′ and R: 5′-CCA GTT GTT GGT AAC AAT GCC ATG-3′.

The number of samples used was *n* = 6 per group. One point was removed from the NCX1 data for SSPN^−/−^ controls as it was identified as an outlier by Prism Graphpad using the ROUT (Q = 1%) method to facilitate plotting but did not alter the results, with nonsignificant trends between the groups. The SSPN^−/−^ IR injury NCX1 consists of *n* = 6; however, one point is obscured by a similar value.

## 8. Hemodynamic Recording in Mice

The mice were anesthetized under 2% Isoflurane with 100% O_2_ for 10 min while electrocardiography (ECG) traces were recorded for a period of 5 min while at rest and then after stress induction (Isoproterenol; β-adrenergic receptor agonist, 3 mg/kg) for 5 min. The ECG recording and analysis were performed using Powerlab and Lab chart (AD Instrument, Colorado Springs, CO, USA).

## 9. Cardiac Ischemia-Reperfusion (IR) Injury In Situ

Mice were euthanized under isoflurane anesthesia and the heart was removed after performing a thoracotomy. The aorta was then cannulated in the Langendorff mode, which involved retrograde perfusion of the aorta at 37 °C with a perfusion buffer containing (mM) 139 NaCl, 4 KCl, 14 NaHCO_3_, 1.2 NaH_2_P0_4_, 1 MgCl_2_, 1.5 CaCl_2_ and 10 glucose as previously described [47,48]. The solution was filtered prior to use and oxygenated with carbon dioxide (95% O_2_/5% CO_2_) to achieve a pH of 7.4. The heart was allowed to equilibrate for 10 min prior to initiation of the ischemia-reperfusion (IR) protocol. The baseline ECG measurements were collected for ten minutes prior to exposing the heart to ischemic conditions (no flow) for 30 min and the ECG readings were recorded during continuous reperfusion for an additional 60 min. After completion of the experiment, the hearts were used for protein, mRNA and histological (infarct size) analyses.

## 10. Histology and Infarct Size Measurement

Histological analysis of the heart tissues was performed to measure infarct size. To this end the heart tissues were embedded horizontally in OCT and flash-frozen in liquid N_2_-cooled 2-methylbutane [49]. Next, the hearts were wrapped in clear plastic to prevent drying and partially thawed at −20 °C and sectioned into 1 mm sections and stained with tetrazolium chloride (TTC). To measure infarct size the stained heart slices were photographed using a high-resolution camera, and the images were analyzed using ImageJ. The color threshold mode was utilized to differentiate between the infarcted and viable tissue, and the pictures were adjusted for hue, saturation, and brightness to enhance visualization of the pale infarcted regions [50]. The select/analyze/measure options in ImageJ were used to count the number of encircled pixels in the infarct zone. A 2 cm standard was utilized to convert the number of pixels to the actual area in mm^2^. For each heart, three adjacent sections were used to calculate the infarct volume, and the number of pixels in the infarct zones were counted to obtain the infarct size in mm^2^ for each heart.

## 11. Myocyte Isolation and Ca^2+^ Handling Measurements

Myocytes were isolated using a modified collagenase/protease method [51,52,53], and all the experiments were conducted in Tyrode’s solution containing (mM) levels of 2 CaCl, 134 NaCl, 5.4 KCl, 1 MgCl_2_, 10 glucose, and 10 HEPES (pH 7.4) at room temperature. Intracellular Ca^2+^ regulation was measured using the previously described protocol [53]. Fura-2 AM-loaded cardiomyocytes were field-stimulated at a frequency of 1 Hz for a 10 s duration to obtain baseline measurements, followed by a 40 s exposure to isoproterenol (1 μM) or vehicle. The SR Ca^2+^ content was then quantified by the application of caffeine (10 mM, 5 s). The Ca^2+^ fluorescence ratios (F_ratio_) were recorded and normalized with an average vehicle group. The data were analyzed using the commercially available data analysis software IonWizardTM 6.6, IonOptix, Milton, MA, USA.

## 12. Western Blotting

Heart samples were lysed at 4 °C using a urea-based extraction buffer [8]. After quantification of the lysates, using a Bradford protein assay, Laemmli buffer was added to each sample. Next 25 µg of protein were loaded onto a 10% SDS-polyacrylamide gel and size-fractionated by electrophoresis at a constant current. The proteins were transferred via electrophoresis onto polyvinylidene difluoride (PVDF) membranes at 5 V/cm for 16–20 h at 4 °C. After incubation in blocking solution (PBS-T, 1% bovine serum albumin) for 1 h, membranes were washed three times in TBS buffer (50 mmol/L NaCl, 10 mmol/L Tris, pH 7.0, 1 mmol/L EDTA, 0.1% Tween-20) before incubation with the primary antibody (1:1000, ANP, NOX4, oxidized CaMKII, GAPDH, purchased from Fisher Scientific, Hampton, NH, USA). Antibody binding was detected using the Enhanced Chemiluminescence (ECL) method according to the manufacturer’s instructions (Amersham Corp., Arlington Heights, IL, USA). The Western blot data showed results of the proteins plotted on the same membrane and signals reported at the same exposure; therefore, the same GAPDH blot is shown for all the proteins probed.

## 13. Reactive Oxygen Species Detection

Isolated cardiomyocytes were loaded with DCFDA/H2DCFDA (3 μM) for 15 min. After they were washed three times with dye. The fluorescent signals were imaged using fluorescence microscopy at baseline for up to 60 min. The fluorescent intensity of the captured images was measured using ImageJ 1.53k. 

## Figures and Tables

**Figure 1 ijms-24-11868-f001:**
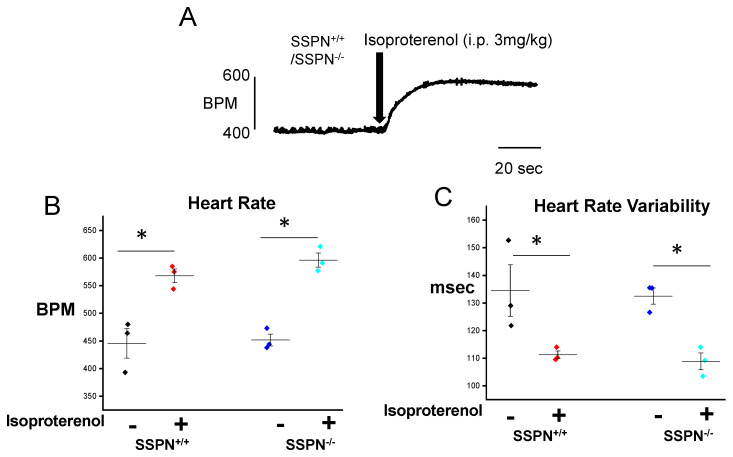
Hemodynamic response of sarcospan-deleted mice after stress testing. (**A**) Representative electrocardiogram (ECG) traces; (**B**) heart rate, beats per minute (HR, BPM) baseline (5 min) vs. after intraperitoneal i.p. isoproterenol (Iso) injection (5 min, 3 mg/kg); (**C**) heart rate variability (HRV, ms) baseline vs. after Iso injection. *n* = 3 per group. Black symbols indicate pre-treatment SSPN^+/+^, red symbols depict post-treatment SSPN^+/+^, royal blue symbols indicate pre-treatment SSPN^−/−^, and light blue symbols depict post-treatment SSPN^−/−^ results. Statistics were performed using Student *t*-test. Significance is indicated as * *p* < 0.05.

**Figure 2 ijms-24-11868-f002:**
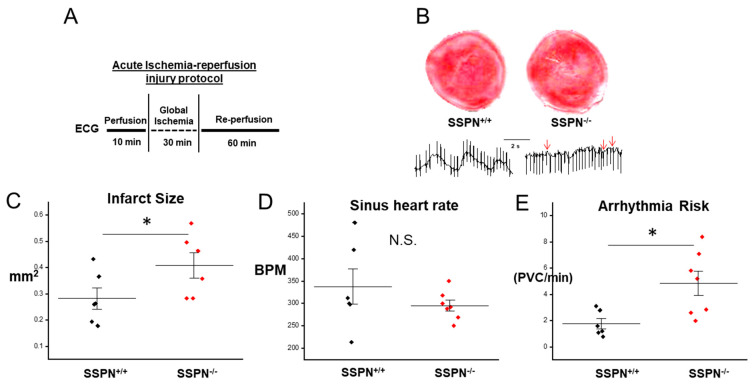
Deletion of sarcospan in intact hearts increases the risk of arrhythmias after ischemia-reperfusion injury. (**A**) A schematic is shown that describes the acute ischemia-reperfusion (IR) procedure in ex vivo hearts; (**B**) representative images of infarct size and electrocardiogram (ECG) traces in SSPN^+/+^ and SSPN^−/−^ hearts subjected to cardiac IR injury. Red arrows indicate: premature ventricular contraction (PVC); (**C**) myocardial infarct size is reported as mm^2^ (*n* = 3 per group, two sections per mouse); (**D**) sinus heart rate measurements during resting 10 min perfusion are shown as beats per minute (BPM); (**E**) arrhythmia risk of SSPN^+/+^ and SSPN^−/−^ hearts are assessed using electrocardiogram (ECG) by counting the number of pre-ventricular contractions (PVCs) per minute after IR injury (*n* = 6–7 per group). Black symbols indicate pre-IR SSPN^+/+^, red symbols depict SSPN^−/−^, Statistics were performed using the Student *t*-test and significance is indicated as * *p* < 0.05 and non-significant comparisons as N.S.

**Figure 3 ijms-24-11868-f003:**
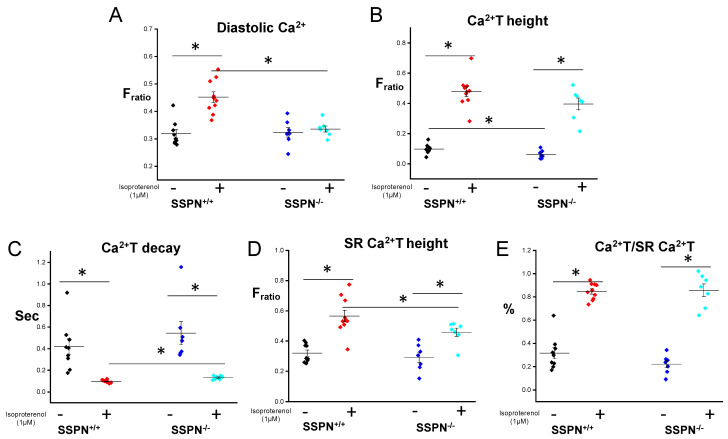
Intracellular Ca^2+^ handling is altered in sarcospan-deficient cardiomyocytes. Fura-2 AM-loaded cardiomyocytes were field-stimulated (1 Hz, 10 s) to obtain data concerning baseline and 40 s exposure to isoproterenol (1μM). To determine SR Ca^2+^ content cardiomyocytes were exposed to caffeine (10 mM, 5 s). (**A**) Average data for diastolic Ca^2+^ levels reported as Fratio; (**B**) average data for systolic Ca^2+^ transient height (release); (**C**) average data for Ca^2+^ height decay; (**D**) average data for caffeine induced Ca^2+^ transient height, which estimates SR Ca^2+^ contents; (**E**) average data for Ca^2+^ transient height normalized to caffeine induced Ca^2+^ transient height (fractional release (%), which estimates functional SR Ca^2+^ release. (*n* = 7–10 cells per group). Black symbols indicate pre-treatment SSPN^+/+^, red symbols depict post-treatment SSPN^+/+^, royal blue symbols indicate pre-treatment SSPN^−/−^, and light blue symbols depict post-treatment SSPN^−/−^ results * vs. *p* < 0.05. Statistics were performed using the Student *t*-test. Significance is indicated as * *p* < 0.05.

**Figure 4 ijms-24-11868-f004:**
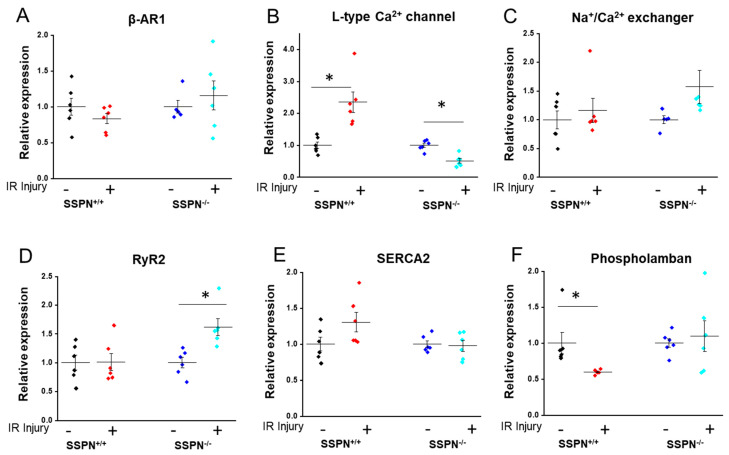
Rapid gene expression changes in SSPN-deficient hearts occur after acute ischemia-reperfusion injury. Rapidly induced epigenetic changes in response to acute IR were measured using qRT-PCR in heart tissue after ischemia-reperfusion injury in a perfused heart setting. Relative mRNA levels were normalized to β-actin expression and the group control average. In (**A**–**C**) averages of plasma membrane ion channels are shown; whereas in (**D**–**F**) averages of sarcoplasmic reticulum ion channels are shown. *n* = 6 hearts per group. Black symbols indicate pre-treatment SSPN^+/+^, red symbols depict post-treatment SSPN^+/+^, royal blue symbols indicate pre-treatment SSPN^−/−^, and light blue symbols depict post-treatment SSPN^−/−^ results * vs. *p* < 0.05. Statistics were performed using the Student *t*-test, separately comparing SSPN^+/+^ and SSPN^−/−^ groups. Significance is indicated as * *p* < 0.05.

**Figure 5 ijms-24-11868-f005:**
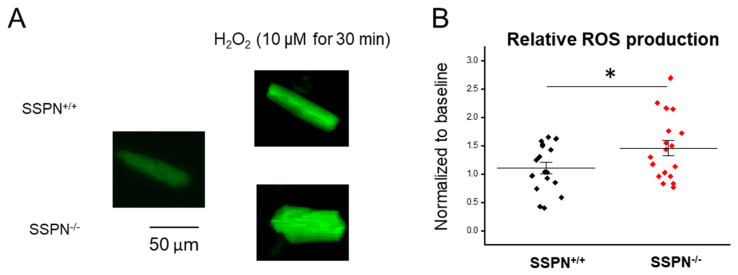
Hydrogen peroxide-induced reactive oxygen species (ROS) production in cardiomyocytes isolated from sarcospan-deleted mice. Cardiomyocytes were loaded DCFDA/H2DCFDA (3 μM) for 15 min, then imaged using fluorescent signals at baseline for up to 60 min. The average incubation time of H_2_O_2_ incubation was 30 min per group. (**A**) Representative florescent cardiomyocyte images for baseline and chronic H_2_O_2_ exposure; (**B**) average date of ROS production. *n* = 17–18 cardiomyocytes per group. Black symbols indicate SSPN^+/+^ and red symbols depict SSPN^−/−^. Statistics were performed using the Student *t*-test. Significance is indicated as * *p* < 0.05.

**Figure 6 ijms-24-11868-f006:**
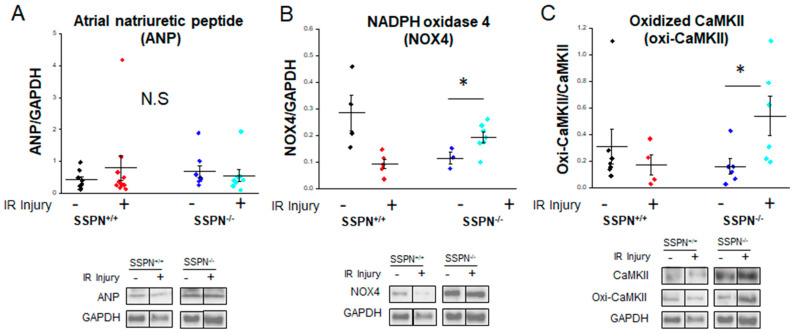
Alterations in expression of stress responsive proteins in sarcospan-deficient hearts after ischemia-reperfusion injury. (**A**) Average data for atrial natriuretic peptide (ANP) expression; (**B**) average data for NADPH oxidase 4 (NOX4) expression; (**C**) average data for oxidized CAMKII expression, it was normalized by CAMKII expression. Note that CAMKII expression was not different among the groups (*n* = 3–10 hearts per group). Black symbols indicate pre-treatment SSPN^+/+^, red symbols depict post-treatment SSPN^+/+^, royal blue symbols indicate pre-treatment SSPN^−/−^, and light blue symbols depict post-treatment SSPN^−/−^ results * vs. *p* < 0.05. Statistics were performed using the Student *t*-test. Significance is indicated as * *p* < 0.05.

**Table 1 ijms-24-11868-t001:** **Echocardiography analysis of SSPN^+/+^ and SSPN^−/−^ mice.** Both male and female mice were evaluated by echocardiography. Values are averages and errors are shown as S.E.M. Student’s *t*-test were performed to detect statistical differences and considered significant if *p* < 0.05 and marked with an *. Abbreviations are as follows: HR = heart rate, LV FS (%) left ventricular fractional Shortening, IVCT isovolumic contraction time (msec), IVRT isovolumic relaxation time (msec), E/A ratio.

*Echo Parameter*	*SSPN^+/+^ (n = 5)*	*SSPN^−/−^ (n = 8)*
*HR* (bpm)	491.6 ± 95.6	475.5 ± 27.2
*LV FS* (%)	38.46 ± 2.20	37.59 ± 1.21
*IVCT* (ms)	16.61 ± 3.06	15.03 ± 1.38
*IVRT* (ms)	18.34 ± 0.96	20.28 ± 1.94
*E/A*	1.233 ± 0.20	1.487 ± 0.56

## Data Availability

All data obtained in this study are provided in this manuscript.

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
