# Peer review of "Sarcospan Deficiency Increases Oxidative Stress and Arrhythmias in Hearts after Acute Ischemia-Reperfusion Injury"

_ijms, 2023, doi:10.3390/ijms241411868_

Round 1

Reviewer 1 Report

Please address the following issues:

Material and method

·        The number of mice is not clarified per each group.

·        The author should explain the wide range of age of mice (4-8 months ) why???

·        Please provide ethical approval number

·        in Western blot section please revise what is meant by (REF)

Results

·        Please revise “ To better understand the function al SR Ca2+ release, “

·        Western blot figures need to be changed to a better resolution one 

Author Response

We thank the reviewers for considering our manuscript for publication in the International Journal of Molecular Sciences, Special Issue: Structural, Chemical, and Energetic Signals in Striated Muscle Function. We have revised the manuscript according to the reviewers’ concerns. A point-by-point summary of the revisions is provided in response to the comments. We believe that the remaining concerns have been readily addressed. All of the changes in the revised MS are indicated in red font.

Reviewer 2 Report

In the current study, Hwang and colleagues tested the role of the integrin-associated protein, SSPN, in ruling cardiomyocyte Ca2+ handling and response to IR injury. The primary conclusion drawn by the Authors is that increased arrhythmia susceptibility in SSPN-/- hearts post-IR injury can stem from perturbed alterations in Ca2+ handling and lower capacity to offset oxidative stress.

From a pathophysiological point of view, the original hypothesis is intriguing and potentially relevant. And there are some good data in the manuscript. However, as presented, the study is a collage of interesting observations missing a clear mechanistic fil-rouge/coherence. Moreover, there are several concerns regarding the experimental approaches and the interpretation of the data. Some findings are left without any explanation.

Major Concerns

1)      Results. The cardiac phenotype of SSPN-/- mice is poorly characterized. Fig.1 depicts ECG traces and HRV; however, it is not clear whether, in these mice, there are other fundamental changes, such as modifications in myocardial contractility and relaxation. At least, echo data should be included.

2)      Results. Data in Fig.3. I’m not sure how to interpret the following statements/data: “Baseline SSPN+/+ and SSPN-/- cardiomyocytes exhibited similar F ratios indicating similar levels of diastolic Ca2+, however after acute Iso exposure SSPN+/+ cardiomyocytes the F ratio significantly increased indicating an increase in diastolic Ca2+ levels, however, it remained unchanged in SSPN-/- cardiomyocytes indicating an anergic response to β-adrenergic stimulation.” Why defining this response “anergic” when functional properties, such sarcomere shortening were not measured?

3)      If the Authors imply that knocking down SSPN may lead to increased oxidative stress (either under stimulating or basal conditions), then one would expect the protein involved in Ca2+ cycling to be oxidatively altered. And the Authors advance this possibility in the Discussion. So may questions are: 1) should the increased ROS production following ISO reduce the overall Ca2+ transient in SSPN mice?

4)      In the same vein, data presented in Fig. 4 do not tell us whether any or all of the proteins examined is modified, and how, after oxidative stress.

5)      Why PLN expression decreases in SSPN+/+ but not SSPN-/- mice. These data are left without any possible explanation.

6)      What was the rationale beyond studies with H2O2, i.e., Fig.5? It seems that H2O2 treatment if “thickening” the myocytes in SSPN-/- mice but not SSPN+/+

7)      Data included in Fig.6 are interesting but lacking a coherent rationale. Moreover, ANP is typically used as a marker of hypertrophy. Is the putative increase in ROS generation is SSPN-/- mice accompanied by a higher decrease of apoptosis?

8)      The upregulation of NADPH oxidase 4 calls for a significant alteration of mitochondrial function. However, the status of these organelles has not been evaluated here.

9)      What the oxidation of CaMKII can lead to in terms of cardiac structure and function?

Minor Concerns

10)   The Introduction is a bit too long. It should be kept around the “topics” effectively touched upon in the study.

11)   Including some original ECG/PVC traces would enrich the manuscript.  

Author Response

(The authors gave the same response as above.)
